# Kinase Inhibitors in Genetic Diseases

**DOI:** 10.3390/ijms24065276

**Published:** 2023-03-09

**Authors:** Lucia D’Antona, Rosario Amato, Carolina Brescia, Valentina Rocca, Emma Colao, Rodolfo Iuliano, Bonnie L. Blazer-Yost, Nicola Perrotti

**Affiliations:** 1Department of Health Sciences, University “Magna Graecia” at Catanzaro, 88100 Catanzaro, Italy; 2Medical Genetics Unit, University Hospital “Mater Domini” at Catanzaro, 88100 Catanzaro, Italy; 3Department of Experimental and Clinical Medicine, University “Magna Graecia” at Catanzaro, 88100 Catanzaro, Italy; 4Department of Biology, Indiana University Purdue University, Indianapolis, IN 46202, USA

**Keywords:** kinase inhibitors, genetics, RAS pathway, mTOR pathway, Wnt pathway

## Abstract

Over the years, several studies have shown that kinase-regulated signaling pathways are involved in the development of rare genetic diseases. The study of the mechanisms underlying the onset of these diseases has opened a possible way for the development of targeted therapies using particular kinase inhibitors. Some of these are currently used to treat other diseases, such as cancer. This review aims to describe the possibilities of using kinase inhibitors in genetic pathologies such as tuberous sclerosis, RASopathies, and ciliopathies, describing the various pathways involved and the possible targets already identified or currently under study.

## 1. Introduction

Gene variants may have functional consequences on the protein product. The variants can be classified as loss of function (LOF) when the protein function is reduced or lost and gain of function (GOF) when the protein function is enhanced or a new function is acquired [1].

The observation of a close similarity between growth factor receptors endowed with a tyrosine kinase enzymatic activity and virally encoded oncogenes [2,3] introduced the general idea that gain of function somatic mutations in genes coding for endogenous kinases involved in the regulation of cell proliferation may be responsible of the neoplastic phenotype [4,5]. A review of this topic is beyond the scope of the present article, but a brief mention to the role of Abl, EGF/HER2, RAF and other kinases in human tumors can be sufficient to underline the statement [6,7,8,9].

Based on these observations, kinase inhibitors have been used for many years in tumor therapy, and new monoclonal antibodies—as well as small inhibitory molecules with different indications in tumor therapy [10,11,12]—are approved every year by the regulatory agencies. HER2 inhibitors are currently used in HER2-positive carcinomas, including breast, colon, and non-small-cell lung (NSCLC) cancers, Abl inhibitors are used in chronic myeloid leukemia. The list is long and is continuously expanding [13,14,15,16,17,18,19,20,21].

More recently kinase activation has been considered in the understanding of the mechanisms underlying the development of the clinical phenotype in rare genetic diseases. A germline mutation in a gene coding either for a kinase or for one of the kinase regulators can indeed be responsible for diseases, thus opening to the possibility of a targeted therapy of otherwise untreatable diseases. In this review, we will focus on some genetic diseases that seem to be associated with the activation of kinase-regulated pathways. In some of these diseases treatment with kinase inhibitors has been recently introduced or at least proposed. In particular we will discuss the use of kinase inhibitors in tuberous sclerosis, in RASopathies, and in ciliopathies associated with either the mTOR, the RAS, or the Wnt pathway, respectively.

## 2. Tuberous Sclerosis Complex

TSC is a disease characterized by autosomal dominant inheritance and has an incidence of 1 in 6000–10,000 individuals [22]. It is involved in the development of very different pathologies that involve the brain, heart, kidneys, and skin. 

### 2.1. Clinical Manifestation

Neurological manifestations represent the most important cause of impairment in most patients. These include epilepsy, present in 90% of patients [23]; cortical tubers; subependymal nodules; giant cell astrocytomas; intellectual disability; autistic spectrum disorder; and behavioral problems [24,25].

Most patients with epilepsy have seizures in the first year of age [26]. Some studies have speculated that the seizures originate from cortical tubers, but the mechanism is still being studied [27]. Long-term damage depends on the age of onset and the severity of seizures [28,29].

The cortical tuber, from which the disease takes its name, is a focal malformation present in 80–90% of patients. It is due to a failure of cell differentiation with neuronal migration during neurological development [30].

Renal lesions are very common in TSC, the most frequent being angiomyolipoma, a hamartoma composed of adipose tissue, blood vessels, and smooth muscle [31]. Although hamartoma is benign, it often bleeds and can lead to renal failure [32]. Renal cysts, combined with angiomyolipomas (AMLs), are also very common and suggest the diagnosis of TSC. Dermatologically, hypomelanotic macules or ash leaf spots can be observed in 90% of patients and are observed from birth or in early infancy. At 9 years of age, angiofibromas and sebaceous adenomas appear in 70% of patients [33,34]. Cardiac manifestations include rhabdomyoma, which can be as early as the 22nd week of gestation [35] but often tend to regress by the third year of age. Pulmonary involvement is less common, although lymphangioleiomyomatosis is sometimes seen only in adult females [36].

### 2.2. Genetics of TSC

TSC is caused by mutations in two genes, the TSC1 gene located on chromosome 9 (9q34), which encodes the hamartina protein, and TSC2 located on chromosome 16 (16p13.3), which encodes the tuberin protein [37,38]. TSC1 mutations have been identified in ~10–20% of patients, whereas TSC2 mutations have been identified in ~70–90% of patients clinically diagnosed with TSC [39,40]. The proteins produced by these two genes physically interact with high affinity to form a heterotrimeric complex, termed the TSC protein complex, with the TBC1 domain family member 7 (TBC1D7), [41,42] and act on the signaling pathway of mTORC1 (rapamycin complex 1), a serine/threonine kinase involved in many cellular processes, such as cell growth, proliferation, and response to extracellular stress [43,44,45]. The major driver of the cellular hyperplasia and tissue dysplasia seen in TSC is the overactivation of the mTORC1 signaling pathway. Under normal conditions, growth factors stimulate the PI3K (phosphatidylinositol 3-kinase) and Ras–MAPK (mitogen-activated protein kinase) pathways, thereby inhibiting the TSC protein complex and activating mTOR signaling (Figure 1) [46,47]. mTOR acts through two complexes, mTORC1 and mTORC2. The loss of function of TSC genes results in an increase in mTORC1 and a decrease in mTORC2 signaling [48]. The mechanism by which this occurs is not clear; it is probably due to loss of direct binding of the amartin/tuberin complex to mTORC2. Alternatively, increasing mTORC1 activates a negative feedback loop from p70S6K inhibiting IRS-1 by inhibiting the PI3-kinase-dependent activation of mTORC2.

In mTORC1, mTOR forms a complex with raptor and GβL 1, 2, 3 proteins and acts in response to extracellular stimuli by phosphorylating two distinct molecules: ribosome S6 protein kinase (p70S6K)—which in turn phosphorylates ribosomal protein S6, which leads to the recruitment of ribosomes with an increase in protein synthesis essential for cell growth [23]—and 4E-binding protein1 (4EBP1), whose activation induces the translation of mRNAs that support tumor cells when they are starved [49,50]. mTORC1 can be activated by Rheb (a RAS homolog GTP binding protein enriched in the brain). Under normal conditions, the function of the TSC complex is to inhibit this activation. When there are mutations that prevent the correct formation of the complex [51], constitutively active Rheb overactivates the mTORC pathway with consequent metabolic reprogramming characterized by an increase in the synthesis of nucleotides, lipids, and proteins [41,42,52]. In mTORC2, mTOR forms a complex with rictor (mAVO3), SIN1, and mLST8 (GBL) and acts in response to growth factors and PI3K by phosphorylating AGC kinases, SGK1, and AKT, regulating cell proliferation [53].

In the central nervous system, the mTOR pathway is involved in neuronal maturation processes such as migration, cortical lamination, and dendritic arborization [54]. The loss of function of the TSC1/TSC2 complex deregulates these processes with consequent anomalies in development and neuronal activity, which may be responsible for epilepsy and TSC-associated neuropsychiatric disorders (TANDs). Specifically, mTOR activation leads to increased dendritic protein synthesis, with long-term depression mediated by alterations in the postsynaptic glutamate receptor [55,56] also leading to memory deficits in patients with autism spectrum disorders. The mTor pathway also seems to be responsible for skin hypopigmentation that can be explained on the basis of several theories. One of these proposes that the inhibition of mTOR leads to an increase in melanogenesis through an increase at the transcriptional level of enzymes involved in this process [57]. Considering this scientific evidence, it was hypothesized that inhibiting mTOR could reverse many of the clinical findings present in patients with TSC.

### 2.3. Inhibition of mTOR

The mTOR inhibitors in use today are sirolimus and everolimus. The former is also known as rapamycin and has been obtained from soil samples from Easter Island [58]. Everolimus is a sirolimus derivative obtained via the addition of an ethyl ester group. Both compounds induce allosteric dissociation of the cofactor rapTOR (TOR regulation-related protein) that is essential for mTOR function [59]. TSC-associated epilepsy does not respond well to treatment with traditional antiepileptics. Some trials show that mTOR inhibition can reduce the frequency of epileptic manifestations and reduce the volume of tumors and nodules [60,61]. In a phase III study (EXIST-3) the effect of everolimus was evaluated in 366 participants divided into three groups and treated with placebo, low dose everolimus, and high dose everolimus. The rate limiting parameter is a >−50% reduction in epileptic frequency, and it was seen that this parameter was achieved more frequently (40%) in patients treated with high doses of everolimus and that this effect could be achieved by treating patients with low doses for longer time [62]. This important trial has favored the FDA’s approval of everolimus as an adjuvant to normal therapies, and clinical practice has led to using the drug alone in some patients with TSC. 

A study on rapamycin in patients with TSC-associated Lymphangioleiomyomatosis showed that this drug can lead to a better spirometric performance and a partial relief from persistent “air trapping”, a condition in which there is an accumulation of air in the lungs with difficulty exhaling it [63]. In the same study, it was shown that patients with moderate AMLs have a temporary amelioration of lung function. Lifelong treatment is recommended in order to obtain more stable results [64]. Based on these studies in 2015, the FDA approved the use of rapamycin in patients with AMLs. The use of mTOR inhibitors was evaluated in pediatric patients for the treatment of rhabdomyoma [65]. The study showed that a reduction in symptom severity was reported by 90.9% of treated patients and that a reduction in tumor size was observed in 95.1% of cases. The role of mTOR inhibition remains unclear since remission of this pathology is often spontaneous. However, a recent study, showed that postnatal treatment with the mTOR inhibitor everolimus initiated significant regression of a prenatally diagnosed giant rhabdomyoma of the hearth suggesting a role for mTOR inhibitors in early onset rhabdomyomas [66]. 

Several studies evaluated the effect of mTOR inhibition in the treatment of renal angiomyolipomas. A reduction in the size of the tumor was detected in a 19-year-old patient who received rapamycin. Notable resumption of tumor growth was observed upon discontinuation of the treatment [67]. A phase III study (EXIST-2) started after the discovery of everolimus. The angiomyolipoma response rate was 42% for everolimus and 0% for placebo, with an acceptable safety profile [68]. Similarly, rapamycin treatment was associated with reductions in cyst number, sum diameter, and volume in patients with TSC-associated renal cysts [61,69]. Angiofibromas and disfiguring facial lesions are common patients with TSC. In these patients, the systemic use of mTOR inhibitors can provide a partial response but is associated with serious adverse events. On the other hand, the application of low-dose topical rapamycin decreased the appearance of facial angiofibromas in patients with tuberous sclerosis complex [61].

Several studies are underway to test new inhibitors of the mTOR pathway on tumor models. In the future, some of these could be approved for use on TSC.

The United States Food and Drug Administration has approved temsirolimus to treat advanced renal cell carcinoma. Some researchers have combined the properties of the first- and second-generation inhibitors, creating rapalink-1, a third-generation mTOR inhibitor [70] that is able to inhibit two targets on the mTOR enzyme at the same time. In animal models, another compound, VS-5584, was recently tested and was shown to inhibit both mTOR and PI3K in melanoma cells.

In fact, its oral administration in nude mice suppresses the growth of the A375 melanoma xenograft [71,72,73]. Another compound under study in models of melanoma is SKLB-M8, a millepaquine derivative, which appears to block cell proliferation via the AKT/mTOR pathway and appears to down-regulate angiogenesis by regulating ERK1/2 phosphorylation [74]. In conclusion, a beneficial role of mTOR inhibition has been documented on the quality of life of patients with various TSC related diseases.

## 3. RASopathies

RASopathies affect about 1 in 1000 individuals and are part of a heterogeneous group of diseases in which there is a germline mutation in the genes that code for components and regulators of the Ras/MAPK Pathway (Figure 2). This is one of the best-studied and characterized pathways and is involved in the regulation of growth, cell cycle differentiation, and normal mammalian development.

### 3.1. Clinical Manifestation

These pathologies each have their own peculiar phenotype but have common characteristics, such as cardiac malformations, craniofacial dysmorphologies, skin, ocular and musculoskeletal abnormalities, hypotonia, and predisposition to tumors [75].

### 3.2. Neurofibromatosis 

The first to be identified was Neurofibromatosis type 1 (NF1). This is an autosomal dominant disease that affects 1 in 3000 individuals and is present in one of the parents in 50% of cases, whereas it is the result of a de novo mutation in the remaining 50%. NF1 is characterized by the development of progressive benign and malignant tumors of the central and peripheral nervous system [76]. Diagnosis is mainly based on the presence of café-au-lait maculae, intertriginous freckling, neurofibromas and plexiform neurofibromas, iris Lisch nodules, osseous dysplasia, optic pathway glioma, and other symptoms common to RASophaties in general. The most severe and frequent clinical manifestations in type 1 neurofibromatosis are plexiform neurofibromas (PNFs), low-grade gliomas, and other peripheral nerve sheath tumors. PNFs occurs in 20–50% of patients with NF1 [77]. These are tumors of the peripheral nerves that are generally benign but can cause complications such as disfigurement and pain [78,79]. PNFs are diagnosed in early childhood and may need to undergo surgical excision, although the procedure can be complicated by factors such as proximity to nerves and wide vascularity.

### 3.3. Noonan Syndrome

Noonan syndrome (NS) is one of the most common among RASopathies, in most cases it is inherited in an autosomal dominant fashion, with a variable expression.74,75 Clinical diagnosis is based on peculiar facial characteristics, short stature, cardiac defects—such as pulmonary stenosis or hypetrophic cardiomyopathy—and variable degrees of development delay. Noonan syndrome clinically overlaps with other RASopathies that are the consequence of genetic defects in the same RAS pathway.

In Noonan syndrome with multiple lentigines (NSML)—formerly known as Leopard syndrome, Costello Syndrome (CS), and Cardio-facio-cutaneous syndrome (CFC)—most of the features of Noonan syndrome are associated with severe mental retardation, severe feeding difficulties, myopathy, facial papillomata, and warts, particularly in the nasolabial area, loose skin on the hands and feet, with hyperkeratotic palms and soles. In capillary malformation–arteriovenous malformation syndrome (CM-AVM) fast-flow vascular malformations, including arteriovenous malformations (AVMs) and arteriovenous fistulas are observed. Finally, in Leugius syndrome, multiple café au lait macules without neurofibromas are observed in association with intertriginous freckling, lipomas, macrocephaly, and learning disabilities.

### 3.4. Genetics of RASopathies 

On the basis of the mutated gene of the Ras/MAPK pathway, RASopathies are classified into different groups (Table 1). Some syndromes are due to the mutation of a single gene in this pathway, while others can arise as a result of the mutation of several genes. 

NF1 is caused by a mutation in the NF1 gene encoding the neurofibromin which is RasGAP, a GTPase-activating protein, that regulates Ras negatively. This mutation results in a loss of protein function followed by increased Ras activity that explains the predisposition to the development of malignant tumors observed in NF1 patients [80]. 

Noonan Syndrome is due to germline pathogenic variants involving genes such as PTPN11, SOS1, KRAS, NRAS, SHOC2, CBL and MAP2K1 [81]. The most frequent mutations are in the PTPN11, SOS1, and KRAS genes. These are generally de novo missense mutations that, once established in the germline, can be inherited in an autosomal dominant fashion. It is generally believed that missense mutations in the PTPN11 gene coding for the SHP-2 tyrosine phosphatase cause a gain of function in the enzymatic activity [82] and that active SHP-2 positively regulates cell growth and differentiation by promoting activation of the Ras–MAPK pathway (Figure 2) [83]. Mutations involving the SOS1 gene, a member of the RAS GEF family important in the conversion of GDP into GTP, lead to the activation of RAS [84,85]. The most frequent mutations of this gene occur in the codons that code for the genes involved in the maintenance of its self-inhibited form, leading to an increase in the activity of SOS1 and a continuous activation of RAS [85]. Other frequent mutations that lead to the onset of this syndrome are those affecting KRAS, which lead to a reduction in the hydrolysis of GTP with accumulation of the active form of KRAS [86,87]. Other rarer mutations, leading to the development of Noonan Syndrome, have been observed in the BRAF, SHOC2, and CBL genes [88,89,90]. 

NSML is much rarer than classic NS syndrome, with an autosomal dominant trait. Again, 85% of cases have a mutation affecting of the PTPN11 gene, with enhanced activity of the RAS pathway [91]. 

CS is an autosomal dominant disease characterized by germline mutations of the HRAS gene, located on chromosome 11p15.5. One of the most common variants is pGly12Ser, which leads to greater activation of HRAS following a reduction in GAP-induced GTPase activity [92,93]. Cardio-facio-cutaneous syndrome (CFC) is an autosomal dominant disease that occurs as a consequence of de novo heterozygous mutations in the BRAF, MAP2K1, MAP2K2 and KRAS genes. BRAF is the most frequently mutated gene, leading to a gain of function with consequent hyperactivation of the RAS/MAPK pathway [94]. Mutations affecting MAP2K1 and MAP2K2, known as MEK1 and MEK2, occur in the self-inhibitory region, leading to continuous activation of the kinases, which phosphorylate and activate ERK1 and ERK2 [95]. 

CM-AVM and Costello syndrome represent two extremely rare RASopathies. In this case, we have a heterozygous mutation with loss of function of the RASA1 gene that encodes p120-RasGAP, which is important for the hydrolysis of GTP [96]. In Leugius syndrome, we have the mutation of the SPRED1 gene, which codes for a particular protein, which acts as a negative regulator of RAS by inhibiting the phosphorylation of RAF [97].

**Table 1 ijms-24-05276-t001:** Genes involved in RASopathies.

Disease	Mutated Genes
Neurofibromatosis type 1 (NF1)	NF1 [98,99,100,101]
Noonan syndrome (NS)	PTPN11 [81,82] SOS1 [84,85] KRAS [86,87] BRAF [88] SHOC2 [89] CBL [90]
Noonan syndrome with multiple lentigines (NSML)	PTPN11 [102]
Costello syndrome (CS)	HRAS [92,93]
Cardio-facio-cutaneous syndorme (CFC)	BRAF [94] MAP2K1(MEK1) [95] MAP2K2(MEK2) [95]
Legius Syndrome	SPRED1 [97]

### 3.5. Inibition of RAS Downstream Targets 

Based on these data, the inhibition of RAS downstream targets, such as MEK, has been proposed as a therapeutic option in RASopathies.

Several phase 2 clinical trials have been conducted using drugs that act on the Ras pathway and others such as pirfenidone (fibroblast inhibitor) [103], tipifarnib (farnesyl transferase inhibitor) [104,105], sirolimus (mTOR inhibitor) [106,107] but none of these has led to benefits that justified their use. A 20% reduction in PNF size has been observed with imatinib (a tyrosine kinase inhibitor) but only in 5% of patients [108] whereas pegylated interferon (a growth inhibitor, antiviral and immunomodulator agent) [109] had some effect only in 14% of patients [110]. Several MEK inhibitors have been tested in clinical trials in adults with refractory cancer. Solumetinib is a specific oral MEK 1/2 inhibitor and has been tested in Phase 1 clinical trials in children with plexiform neurofibromas (PNs) and low-grade gliomas [111]. In these studies, tumors were shown to shrink by 20% in 17 out of 24 children with side effects such as gastrointestinal upset, acneiform rush [112] but without serious toxic effects such as ocular toxicity and heart problems [113]. Phase 2 studies (SPRINT TRIAL) confirmed that 70% of patients treated with Solumetinib had a 20% reduction in pNF with a maximum response after 16 cycles [114]. On the basis of these results, Solumetinib received FDA approval for children 2 years of age and older with NF1 and inoperable and symptomatic PNs. Based on these encouraging data, other MEK inhibitors have been developed and tested in trials.

Trametinib, already approved for adult melanoma, is being tested in children with NF1-related PNs with size reduction by 20% in 46% for children with good tolerability [115]. Mirdametinib has similar results in 16-year-old patients with inoperable pNF45.

Low-grade glioma is frequently present in NF1 [116]. It develops in children under 6 years, involving the optic nerve or chiasm and the hypothalamus [117]. It is a mildly malignant tumor, but due to its localization, it can lead to serious consequences such as loss of sight. The therapy of choice is classical chemotherapy [118], which is characterized by poor efficacy and serious adverse reactions [119]. Several cases report the use of MEK inhibitors for the treatment of this tumor, demonstrating its efficacy [120,121]. In trials in which Solumetinib was tested, partial response was found in 40%, and the majority of patients (96%) had 2 years of progression-free survival with mild side effects [111]. 

For the other RASopathies, several studies are being developed. These pathologies are much less common than NF1, but we have animal models [122]. In cardiofaciocutaneous syndrome, for instance, it has been found that prenatal administration of MEK inhibitor prevents the characteristic phenotype of this pathology [123]. At the same time, in studies on mouse models of Noonan syndrome, the use of MEKi improves embryo survival and heart defects. Encouragement on the use of MEK inhibitors to improve the Noonan syndrome phenotype comes from two infants treated with trametinib, who underwent significant cardiac improvements with regression of myocardial hypertrophy [124].

## 4. Ciliopathies

The term “ciliopathic” comes from the Bardet–Biedl syndrome (BBS). Later, it was observed that the malfunction of the cilia manifests itself with a great variety of characteristics common to different syndromes [125]. The term ciliopathy includes very different pathologies that share a common etiology and pathogenesis in mutations in genes involved in the functioning of the cilia. Cilia are scattered on the apical portion of epithelial cells throughout the body. They perform various functions by regulating the cell cycle, developing and maintaining cell polarity, and conducting mechanosensation. Cilia are divided into two groups, one represented by motile cilia and the other represented by non-motile cilia, also called primary cilia [126]. Motile cilia are found on respiratory epithelial cells, ependymal cells of the cerebrospinal fluid spaces, sperm cells, and embryonic node cells during development. Primary cilia act as an antenna for extracellular information and by converting mechanical or chemical stimuli into electrical signals [127]. For signal conversion, there are many Ca^2+^-permeable channels both on the membrane of the cilia and on the basal body, such as polycystin 2 (PC-2), and member 4 of the cation channel subfamily of the transient receptor potential V (TRPV4) (Figure 3 and Figure 4) [128]. 

### 4.1. Clinical Manifestation

Joubert syndrome is a rare syndrome characterized by hypotonia, ataxia, psychomotor delay, irregular breathing pattern, and oculomotor apraxia [125]. This syndrome is associated with mutation of several genes, as reported in (Table 2). Mutations in 34 genes have been shown to be involved in the development of JS—33 classified as autosomal recessive and one as X-linked.

Meckel–Gruber syndrome (MKS) is phenotypically similar to Joubert syndrome (JBTS) and has a characteristic phenotype with occipital encephalocele and other posterior fossa defects, cystic dysplastic kidneys, hepatic bile duct proliferation, and polydactyly. Senior–Løken syndrome leads to retinitis pigmentosa (RP) and renal disease and infantile onset, with several similarities to Joubert’s syndrome. Orofacial syndrome type 1 is a rare X-linked dominant disorder that leads to spontaneous abortions of male fetuses. Features include developmental defects of the oral cavity, face, and fingers;central nervous system (CNS) abnormalities; and cystic kidney disease [129]. Leber’s congenital amaurosis is characterized by retinal dystrophy which occurs during the first year. A characteristic symptom is Franceschetti’s oculodigital sign, consisting of pricking, pressing, and rubbing the eye [129]. Bardet–Biedl syndrome is characterized by renal malformations, with dysplasia nephrophtisis and cystic tubular disease, [130] polydactyl obesity, hypogonadism, and craniofacial defects [131]. Rhins syndrome is characterized by retinitis pigmentosa, hypopituitarism, nephronophthisis, and Skeletal dysplasia (RHYNS). In a patient with RHYNS syndrome, we identified compound heterozygous variants in the TMEM67 gene involved in the role of cilia resulting in abnormal splicing of exon 13 [132].

**Table 2 ijms-24-05276-t002:** Genes involved in ciliopathies.

Syndrome	Gene Involved
Joubert syndrome	INPP5E [133], ARL13B [134], CC2D2A [135], RPGRIP1L [136], TMEM67 [137], NPHP1 [138], AHI1 [139], CEP290 [140], CXORF5 [141], and TMEM216 [142].
Meckel–Gruber syndrome	MKS1 [143], MKS3 (TMEM67) [144], CEP290 [137], RPGRIP1L [136], CC2D2A [145], and TMEM216 [142]
Senior–Løken syndrome	CEP290 (also known as NPHP6 and MKS4) [140], NPHP1 [146], NPHP3 [147], NPHP4 [148], and NPHP5 (also known as IQCB1) [149].
Type 1 orofacial syndrome	OFD1 [150]
Leber’s congenital amaurosis	GUCY2D [151], RPE65 [152], SPATA7 [153], AIPL1 [154], LCA5 [155], RPGRIPL1 [156], CRX [157], CRB1 [158], IMPD1 [159], RD3 [160], CEP290 [140], NPHP5 [149], and RDH12 [161].
Bardet–Biedl syndrome	BBS1 [162], BBS2 [163], ARL6/BBS3 [164], BBS4 [165], BBS5 [166], MKKS/BBS6 [167], BBS7 [168], TTC8/BBS8 [169], B1/BBS9 [170], BBS10 [171], TRIM32/BBS11 [172], BBS12 [173], MKS1/BBS13 [174], CEP290/BBS14 [174], C2ORF86/FRITZ/BBS15 [175], and SDCCAG8/BBS16 [176]
Alström syndrome	ALMS1 [177]
Jeune asphyxiating thoracic dystrophy	IFT [178]
Rhyns syndrome	TMEM67 [132]

### 4.2. The Genetics of Ciliopathies

Mutations in the gene-encoding proteins involved in the development and signal conversion by cilia lead to developmental and degenerative diseases, which exhibit a broad phenotypic spectrum. The cilia perform their action by mediating key developmental pathways, such as Wnt and Shh signaling (Figure 3 and Figure 4). The binding of Wnt to its receptor complex composed of the Frizzled receptor (Fz) family and low-density lipoprotein-related protein 5/6 (LRP5/6) [179] activates various cascades of intracellular signal transduction, the signal activates disheveled (Dsh/Dvl), a cytoplasmic phosphoprotein, at the level of Dsh. This signal can take different routes; one of these is the canonical route (Wnt/β-catenin-dependent), and another is the non-canonical pathway (β-catenin-independent), which further subdivides into the planar cell polarity pathway and the Wnt/Ca^2+^ pathway (Figure 3) [180].

The canonical pathway of Wnt is characterized by the accumulation and translocation in the nucleus of a protein called β-catenin, which is very important in cell adhesion. When Wnt is not bound to its receptor, cytoplasmic β-catenin is phosphorylated by a complex of destruction consisting of Axin, adenomatosis polyposis coli (APC), protein phosphatase 2A (PP2A), glycogen synthase kinase 3 (GSK3), and casein kinase 1α (CK1α). Phosphorylated β-catenin is recognized for ubiquitination, resulting in proteasomal degradation. Binding of Wnt to its receptor complex induces membrane translocation of a negative regulator of Axin signaling, which binds to a conserved sequence in the cytoplasmic tail of LRP5/6 leading to disruption of the APC/Axin/GSK3 complex [179]. The stabilized β-catenin then translocates into the nucleus, where it enhances the transcription of genes regulating several cellular functions related to the development of a neoplastic phenotype, i.e., inflammation, fibrosis, Warburg effect and metabolism, and circadian rhythms. The transcriptional output of beta catenin is mediated by cyclin-dependent kinases, such as cyclin-dependent kinase 8 (CDK8), 19 module [181,182]. Interestingly, the kinase Sgk1 is both a transcriptional target of beta catenin as well as a positive regulator of beta catenin stabilization and nuclear translocation, creating a loop with the WNT/beta catenin signaling that can be an interesting pharmacological target in conditions for which the canonical pathway is dysregulated [183,184]. The non-canonical planar cell polarity pathway involves two domains of Dsh and DAAM 1 (disheveled-associated morphogenesis activator 1). Active DAAM1, in turn, activates PDZ-RhoGEF through a physical interaction. PDZ-RhoGEF recruits ROCK (rho-associated kinase), probably by activating RhoA, which activates myosin II with cytoskeleton reorganization [175]. Furthermore, Dsh via the DEP domain forms a DAAM1-independent complex with Rac GTPase, stimulating Jun kinase (JNK) activity which mediates actin profiling binding. WNT-dependent and WNT-independent effects are involved in the polarity signaling pathway resulting in an intricate interplay between polarity and adhesion [185,186]. 

Another fate of the non-canonical WNT pathway is the WNT/Ca^2+^ pathway. The role of the pathway is to regulate the release of intracellular Ca^2+^ from the endoplasmic reticulum and to regulate signaling pathways involved in dorsal axis formation, and in cell adhesion, migration and separation in tissues during gastrulation. In this pathway, the binding of WNT to the FZD receptor leads to direct interaction and activation of heterotrimeric G proteins, leading to an increase in intracellular Ca^2+^ concentration. Ca^2+^ release activates calcium/calmodulin-dependent kinase II (CamKII), calcineurin, or protein Ki-nase C [187]. Loss of function mutations in genes coding for proteins involved in this pathway are associated with the enhancement of canonical Wnt pathway followed by stabilization of DVL and beta catenin in the cytoplasm and the nucleus [125]. The increased canonical Wnt/beta catenin activity leads to over-proliferation and dysregulation of development in the above-mentioned organs affected in ciliopathies [188]. 

TMEM67, one of a complex of proteins involved in formation of the primary cilium, may be considered a prototype gene coding for a protein that inhibits canonical Wnt through the repression of the expression of homeoboxtype transcription factors, including the HOX5 gene group [189]. Consistent with a role in non-canonical Wnt signalling, TMEM67 is required for the migration of centrioli towards the apical membrane and for the regulation of actin cytoskeleton remodeling. TMEM67 (TMEM67^−/−^) knock-out animals recapitulate several features of MKS, Joubert syndrome, and Rhyns syndrome in humans and correlate with an aberrant increase in Wnt/β-catenin signaling both in vitro and in vivo [132,190]. 

Hydrocephalus, commonly found in MKS3, can be the consequence of an imbalance in fluid-electrolyte homeostasis, finely regulated by ion transporters like TRPV4, whose expression is controlled by Sgk1. In experiments using an epithelial cell line of the choroid plexus in culture, TRPV4 (transient receptor potential channel 4) was identified as a channel protein which, once activated, allows Ca^2+^ and Na^+^ to enter the cell, causing electrolytic changes and movement of fluid through the cells. TRPV4 is found in the plasma membranes of a wide variety of tissues and organs, including the central nervous system. The channel can be activated by alterations of the osmotic balance (hypotonicity), temperature, mechanical stress (pressure), and inflammatory mediators such as arachidonic acid metabolites, integrating multiple stimuli [191]. TRPV4 affects transport proteins in the brain, such as aquaporin-4 in astrocytes [192] and the Ca^2+^-activated Cl^−^channel, TMEM16A, in primary cultures of CP cells [193]. The influx of calcium through TRPV4 appears to stimulate Ca^2+^-sensitive K+ channels in a tissue-specific manner. Ca^2+^-sensitive K+ channels are divided into three categories: large conductance (BK; KCa1.1), intermediate conductance (IK; KCa3.1; KCNN4), or small conductance (SK1, SK2, SK3; KCa2.1, KCa2. 2, KCa2.3 channels) [194]. In endothelial cells of low-resistance arteries and arterioles, a drop in intraluminal pressure stimulates TRPV4, which causes localized Ca^2+^ sparklets, which activate IK channels and reduce arteriole tone. 

Other genes, such as CEP290, are found to be mutated in various syndromes in which normal function of the cilia is lost, including Meck-el-Gruber syndrome, Joubert syndrome, Senior–Løken, and Bardet–Biedl syndrome [195], altough they appear not to be directly involved in the Wnt or Shh pathway. CEP290 encodes a centrosomal protein involved in the assembly and trafficking of cilia. The mutation usually leads to a loss of function of the protein with heterogeneous consequences, as it interacts with a large number of factors [196], such as retinitis pigmentosa GTPase regulator, nephrocystin-4, and nephrocystin-8 [197].

### 4.3. Inibition of TRPV4 and SGK1 

In a porcine CP epithelial cell line, stimulation of endogenous TRPV4 results in multiphasic ion fluxes, which can be blocked by either of two specific TRPV4 antagonists [198]. Interestingly, in a unique rat model MKS, the *Tmem67* genetic model [191], treatment with TRPV4 antagonists (HC067047 and RN 1734) counteracted the development of the hydrocephalic phenotype in homozygous TMEM67^−/−^ rats. On the other hand, TRPV4 agonist treatment exacerbated the hydrocephalus in the affected animals [199]. Given the peculiar role that Sgk1 seems to maintain in modulating canonical WNT signaling and in channel regulation, it is possible that Sgk1 inhibition can also have a role in reducing some of the phenotypic characteristics of ciliopathyes characterized by a dysregulated Wnt canonical pathway (Figure 4). We recently identified a pyrazolo [3,4-d].pyrimidine compound, named SI113, that inhibits the Sgk1 associated kinase activity. The compound inhibits proliferation and restores chemo sensitivity in several human cancer cell models and is well tolerated when administered to immunocompromised models as preclinical models of human cancers [200,201]. Preliminary data show that SI113 may indeed have an effect on the development of the hydrocephalic phenotype in TMEM67^−/−^ rats [202], suggesting that at least some of the phenotypes observed in MKS3 and Joubert syndrome may be the consequence of Sgk1 hyper-expression and that Sgk1 inhibition may be considered in the treatment of ciliopathies for which no drug therapy has been developed.

## 5. Discussion

In recent years, several protein kinase inhibitors have been developed and tested in experimental and clinical settings for the treatment of human tumors given the role that serine and tyrosine kinases have in cell proliferation and malignant transformation. This review focuses on the possible role of kinase inhibitors in the treatment of symptoms associated with rare genetic diseases characterized by dysregulation of cellular pathways involving kinases and phosphatases. Three classes of rare genetic disorders, TSC, RASopathies, and cilyopathies, are described in detail, and the dysregulated pathways are described, namely the mTOR, the RAS/MAPK, and the Wnt patwhays, respectively. We discuss several studies suggesting possible therapeutic strategies based on the use of kinase inhibitors. Some treatments such as everolimus have already been approved by the FDA for the inhibition of the mTOR pathway and are associated with a significant reduction in the frequency of epilepsy in patients with TSC. In addition to the drugs already approved by the FDA, many other mTOR inhibitors are currently being studied in tumor models and will probably also be used in patients with TSC. In RASopathies, MEK appears as the best pharmacological target. Among several MEK inhibitors tested so far, solumetinib proved to be the most effective and was approved by the FDA for children 2 years of age and older with NF1 and inoperable and symptomatic PNs. Also in this case, given the encouraging results, several studies are underway to test other inhibitors of this pathway, such as trametinib and mirdametinib. Finallym we suggest the use of Sgk1 inhibitors in the treatment of symptoms (hydrocephalus) associated with cilyopathies in which a dysregulation of the canonical and non-canonical Wnt pathways leads to the stabilization and activation of beta catenin.

## 6. Conclusions

In conclusion, scientific research is trying to find possible new therapeutic approaches to be used for the treatment of some rare genetic diseases or to try to counteract—at least in part—the symptoms associated with them, and in this area, great results have been obtained use of kinase inhibitors.

## Figures and Tables

**Figure 1 ijms-24-05276-f001:**
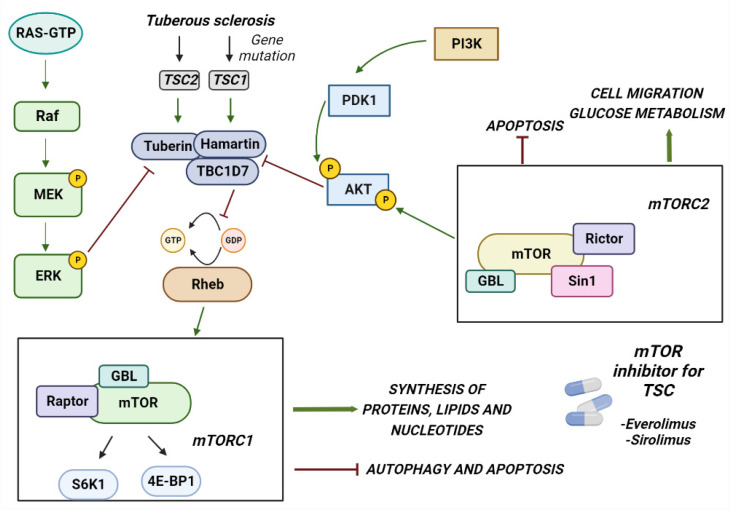
The figure shows how the mTORC1 pathway is involved in the development of TSC. The figure shows the genes involved in the development of TSC and the pathway activated by different proteins. In particular, mTORC1 may contribute to the symptoms of TSC when hyperactivated by the malfunction of the TBC1D7 tuberin and hamartin complex. Two of the FDA-approved kinase inhibitors for the treatment of TSC are also shown in the figure.

**Figure 2 ijms-24-05276-f002:**
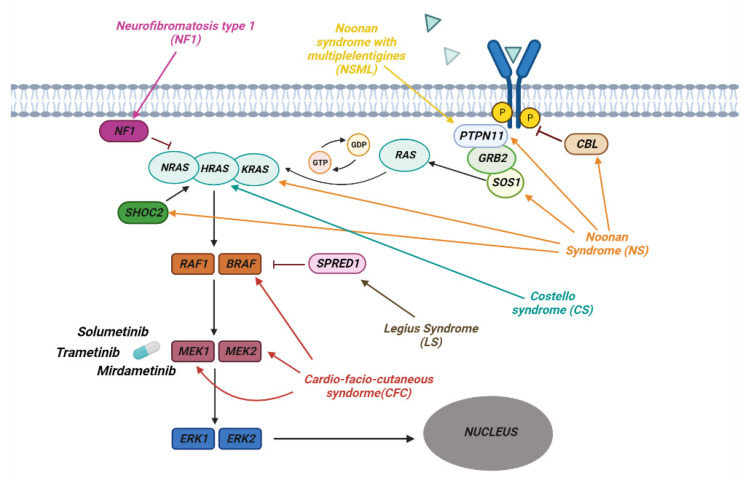
The pathway of RAS/MAPK in RASopathies. The figure shows how the various genes of the Ras/MAPK pathway are involved in the development of the different syndromes associated with RASopathies. Shown are some of the MERK1/2 kinase inhibitors used to treat RASopathies, some of which are FDA-approved.

**Figure 3 ijms-24-05276-f003:**
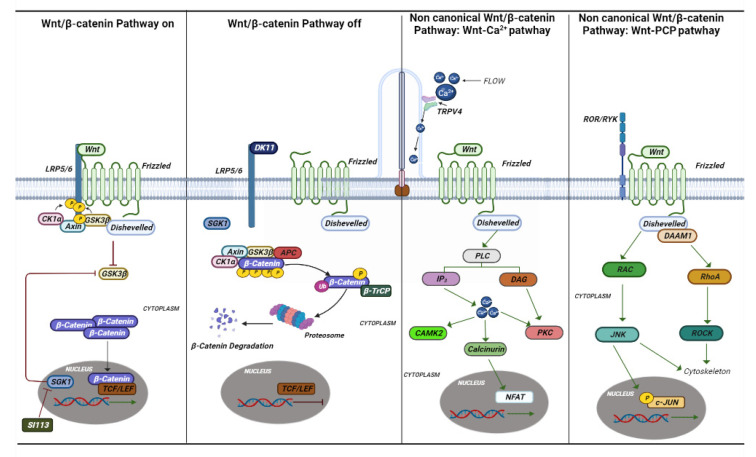
Canonical and non-canonical Wnt pathways. The canonical and noncanonical Wnt pathways are depicted in the figure with the various genes and mechanisms that are activated. In ciliopathies, several studies have shown that this pathway plays an essential role in their onset. Si113 is indicated; an SGK1 kinase inhibitor appears to have an effect on hydrocephalus caused by some ciliopathies.

**Figure 4 ijms-24-05276-f004:**
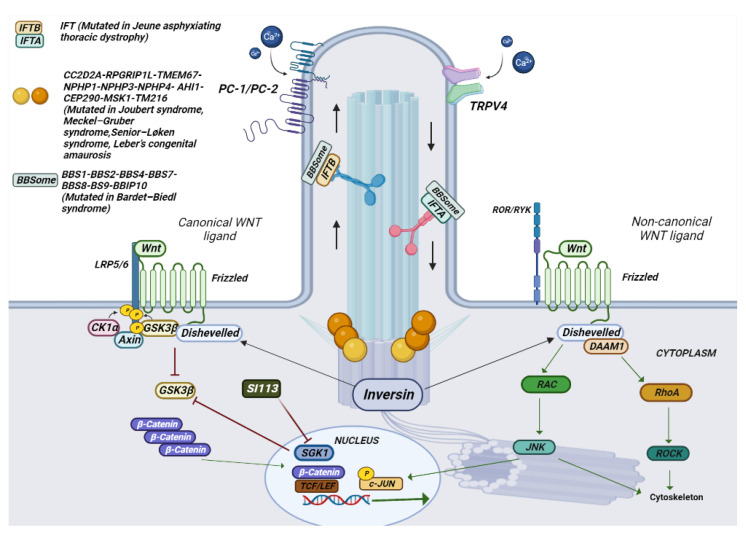
Schematic representation of the composition of a cilium and the various mutated genes in ciliopathies. In the photo it is possible to observe how a cilium is structured in its various components. The different genes mutated in ciliopathies, and their location are represented.

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
