# Peer review of "Kinase Inhibitors in Genetic Diseases"

_ijms, 2023, doi:10.3390/ijms24065276_

Round 1

Reviewer 1 Report

In this manuscript, the authors summarized kinase inhibitors that are involved some genetic diseases, for example, tuberous sclerosis, rasophaties and ciliopathies. Overall, this review is relatively comprehensive, interesting and meaningful. My specific comments are listed below.

1.    The presentation is not very clear. What’s the logicalrelations of the subtitles? To be easier understood, it would be better if the authors can summarize it the types of genetic diseases or by the pathway-related kinase inhibitors?

2.    The authors summarized genetic diseases related genes in the tables. I suggest the related inhibitors should also be included.

3.    It would be better if the authors can add a Discussion section.

Reviewer 2 Report

Genetic disorders are relatively rare but severe congenital conditions that are often difficult to manage in the clinic. While the genes mutated in the disease may not have drugs directly acting on them, kinases are often part of the signaling axis leading to the disorder and can be targeted by various inhibitor compounds. It is extremely valuable to summarize the kinases involved in genetic disorders and how specific kinase inhibitors may facilitate the management of the symptoms.

To facilitate the readers in following this review, it is important to chart out the pathway, especially the epistatic relations between the gene mutation and "druggable" kinases. For the multiple pathways affected in ciliopathy, it would be helpful to make the figure illustrating the components and their regulatory roles. In addition, for the other conditions, the authors have made a table listing the gene mutations, but it would be better to include their effect on the corresponding pathway (activating/inactivating) and examples of kinase inhibitors that may work as remedy.

Reviewer 3 Report

D’Antona et al in their review discuss the use of tyrosine kinase inhibitors as a potential therapeutic solution to treat rare forms of genetically inherited disease, paying special attention to three diseases: tuberous sclerosis (TSC), RASopathies, and ciliopathic diseases.

RTK inhibitors are well known as therapeutic solutions to treat several forms of cancers and this article has good concept to cover the role of these inhibitors in rare genetic diseases. While the concept is well thought, the review is a little hard to follow and lacks a consistent pattern of description that is confusing to read.

The biggest problem is the lack of a consistent pattern in describing each category of the disease. It would be nice if the authors can follow the same pattern of writing for RASopathies and Ciliopathies sections as they have used to describe TSC diseases (following the same sub-headings- description of the disease, clinical manifestations, genetics, and RTK inhibitor drugs in use and under research).

Some other concerns that I have are as follows:

1.       Addition of the role of Rictor and Raptor under normal conditions might help the readers understand why these are targeted to treat the disease. What is the role of mTORC2 in TSC?

2.        Line140, is not clear. What is gas trapping and how does it affect lymhangioleiomyomatosis?

3.       Inclusion of a paragraph about the preclinical and on-going studies in animal models to show new targets being discovered for mTOR pathway might be helpful.

4.       Please check for the correct abbreviation: RASopathies

5.       Fig 3, the description of the “off pathway” is missing in the figure legend.

6.       There is no figure showing the non-canonical Wnt signaling pathway and its role in ciliopathies.

7.       Between lines 305-329, there is no description stating the link between CEP20 mutations (Joubert syndrome) and abnormal Wnt signaling pathway.

8.       Line 380, 383, MKS or MKS3 has not been elaborated but only mentioned in the table. It is not clear how this gene is affecting Wnt signaling.

9.       Lines 383-400 describes the role of TRPV4. Is this an RTK? How is it linked to Wnt signaling?

10.   The Ciliopathy section needs to be reorganized. It is confusing to read and hard to follow. It would help if the different diseases were organized under canonical vs non-canonical Wnt signaling, how mutations in those genes cause abnormal Wnt signaling and how different therapeutic targets are being used today.

Round 2

Reviewer 3 Report

The authors have addressed all my concerns. Inclusion of figures and addition of more description on pathways has greatly helped with the understanding.